# Unsupervised Semantic Segmentation by Distilling Feature Correspondences

**Mark Hamilton**
MIT, Microsoft
markth@mit.edu

**Zhoutong Zhang**
MIT

**Bharath Hariharan**
Cornell University

**Noah Snavely**
Cornell University, Google

**William T. Freeman**
MIT, Google

## Abstract

Unsupervised semantic segmentation aims to discover and localize semantically meaningful categories within image corpora without any form of annotation. To solve this task, algorithms must produce features for every pixel that are both semantically meaningful and compact enough to form distinct clusters. Unlike previous works which achieve this with a single end-to-end framework, we propose to separate feature learning from cluster compactification. Empirically, we show that current unsupervised feature learning frameworks already generate dense features whose correlations are semantically consistent. This observation motivates us to design STEGO (**S**elf-supervised **T**ransformer with **E**nergy-based **G**raph **O**ptimization), a novel framework that distills unsupervised features into high-quality discrete semantic labels. At the core of STEGO is a novel contrastive loss function that encourages features to form compact clusters while preserving their relationships across the corpora. STEGO yields a significant improvement over the prior state of the art, on both the CocoStuff (**+14 mIoU**) and Cityscapes (**+9 mIoU**) semantic segmentation challenges.

## 1 Introduction

Semantic segmentation is the process of classifying each individual pixel of an image into a known ontology. Though semantic segmentation models can detect and delineate objects at a much finer granularity than classification or object detection systems, these systems are hindered by the difficulties of creating labelled training data. In particular, segmenting an image can take over $100\times$ more effort for a human annotator than classifying or drawing bounding boxes (Zlateski et al., 2018). Furthermore, in complex domains such as medicine, biology, or astrophysics, ground-truth segmentation labels may be unknown, ill-defined, or require considerable domain-expertise to provide (Yu et al., 2018).

Recently, several works introduced semantic segmentation systems that could learn from weaker forms of labels such as classes, tags, bounding boxes, scribbles, or point annotations (Ren et al., 2020; Pan et al., 2021; Liu et al., 2020; Bilen et al.). However, comparatively few works take up the challenge of semantic segmentation without *any* form of human supervision or motion cues. Attempts such as Independent Information Clustering (IIC) (Ji et al., 2019) and PiCIE (Cho et al., 2021) aim to learn semantically meaningful features through transformation equivariance, while imposing a clustering step to improve the compactness of the learned features.

In contrast to these previous methods, we utilize pre-trained features from unsupervised feature learning frameworks and focus on distilling them into a compact and discrete structure while preserving their relationships across the image corpora. This is motivated by the observation that correlations between unsupervised features, such as ones learned by DINO (Caron et al., 2021), are already semantically consistent, both within the same image and across image collections.

As a result, we introduce STEGO (**S**elf-supervised **T**ransformer with **E**nergy-based **G**raph **O**ptimization), which is capable of jointly discovering and segmenting objects without human supervision. STEGO distills pretrained unsupervised visual features into semantic clusters using a novel

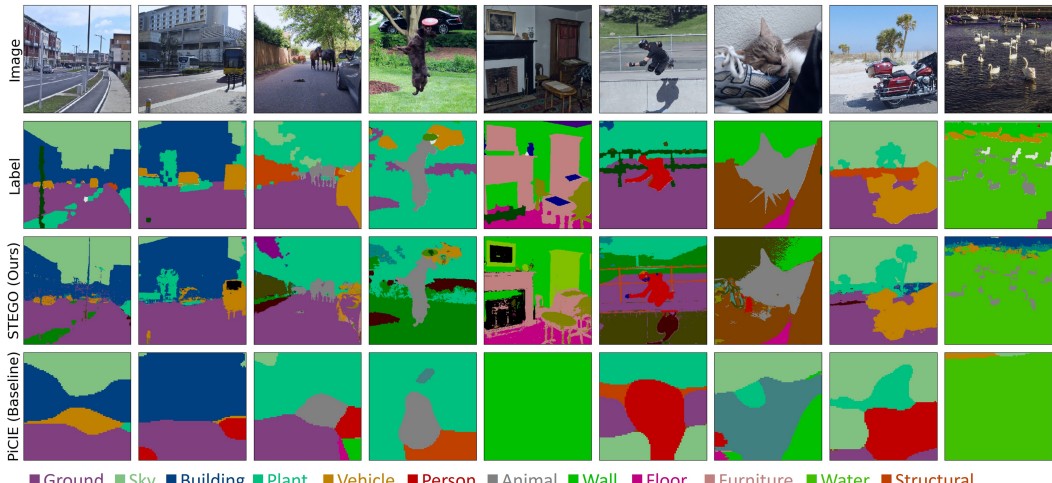

Figure 1: Unsupervised semantic segmentation predictions on the CocoStuff (Caesar et al., 2018) 27 class segmentation challenge. Our method, STEGO, does not use labels to discover and segment consistent objects. Unlike the prior state of the art, PiCIE (Cho et al., 2021), STEGO's predictions are consistent, detailed, and do not omit key objects.

contrastive loss. STEGO dramatically improves over prior art and is a considerable step towards closing the gap with supervised segmentation systems. We include a short video detailing the work at https://aka.ms/stego-video. Specifically, we make the following contributions:

- Show that unsupervised deep network features have correlation patterns that are largely consistent with true semantic labels.
- Introduce STEGO, a novel transformer-based architecture for unsupervised semantic segmentation.
- Demonstrate that STEGO achieves state of the art performance on both the CocoStuff (**+14 mIoU**) and Cityscapes (**+9 mIoU**) segmentation challenges.
- Justify STEGO's design with an ablation study on the CocoStuff dataset.

## 2    RELATED WORK

**Self-supervised Visual Feature Learning**    Learning meaningful visual features without human annotations is a longstanding goal of computer vision. Approaches to this problem often optimize a surrogate task, such as denoising (Vincent et al., 2008), inpainting (Pathak et al., 2016), jigsaw puzzles, colorization (Zhang et al., 2017), rotation prediction (Gidaris et al., 2018), and most recently, contrastive learning over multiple augmentations (Hjelm et al., 2018; Chen et al., 2020a;a;c; Oord et al., 2018). Contrastive learning approaches, whose performance surpass all other surrogate tasks, assume visual features are invariant under a certain set of image augmentation operations. These approaches maximize feature similarities between an image and its augmentations, while minimizing similarity between negative samples, which are usually randomly sampled images. Some notable examples of positive pairs include temporally adjacent images in videos (Oord et al., 2018), image augmentations (Chen et al., 2020a;c), and local crops of a single image (Hjelm et al., 2018). Many works highlight the importance of large numbers of negative samples during training. To this end Wu et al. (2018) propose keeping a memory bank of negative samples and Chen et al. (2020c) propose momentum updates that can efficiently simulate large negative batch sizes. Recently some works have aimed to produce spatially dense feature maps as opposed to a single global vector per image. In this vein, VADeR (Pinheiro et al., 2020) contrasts local per-pixel features based on random compositions of image transformations that induce known correspondences among pixels which act as positive pairs for contrastive training. Instead of trying to learn visual features and clustering from scratch, STEGO treats pretrained self-supervised features as input and is agnostic to the underlying feature extractor. This makes it easy to integrate future advances in self-supervised feature learning into STEGO.

**Unsupervised Semantic Segmentation**   Many unsupervised semantic segmentation approaches use techniques from self-supervised feature learning. IIC (Ji et al., 2019) maximizes mutual information of patch-level cluster assignments between an image and its augmentations. Contrastive Clustering (Li et al., 2020), and SCAN (Van Gansbeke et al., 2020) improve on IIC's image clustering results with supervision from negative samples and nearest neighbors but do not attempt semantic segmentation. PiCIE (Cho et al., 2021) improves on IIC's semantic segmentation results by using invariance to photometric effects and equivariance to geometric transformations as an inductive bias. In PiCIE, a network minimizes the distance between features under different transformations, where the distance is defined by an in-the-loop k-means clustering process. SegSort (Hwang et al., 2019) adopts a different approach. First, SegSort learns good features using superpixels as proxy segmentation maps, then uses Expectation-Maximization to iteratively refine segments over a spherical embedding space. In a similar vein, MaskContrast (Van Gansbeke et al., 2021) achieves promising results on PascalVOC by first using an off-the-shelf saliency model to generate a binary mask for each image. MaskContrast then contrasts learned features within and across the saliency masks. In contrast, our method focuses refining existing pretrained self-supervised visual features to distill their correspondence information and encourage cluster formation. This is similar to the work of Collins et al. (2018) who show that low rank factorization of deep network features can be useful for unsupervised co-segmentation. We are not aware of any previous work that achieves the goal of high-quality, pixel-level unsupervised semantic segmentation on large scale datasets with diverse images.

**Visual Transformers**   Convolutional neural networks (CNNs) have long been state of the art for many computer vision tasks, but the nature of the convolution operator makes it hard to model long-range interactions. To circumvent such shortcomings, Wang et al. (2018); Zhang et al. (2019) use self-attention operations within a CNN to model long range interactions. Transformers (Vaswani et al., 2017), or purely self-attentive networks, have made significant progress in NLP and have recently been used for many computer vision tasks (Dosovitskiy et al., 2020; Touvron et al., 2021; Ranftl et al., 2021; Caron et al., 2021). Visual Transformers (ViT) (Vaswani et al., 2017) apply self-attention mechanisms to image patches and positional embeddings in order to generate features and predictions. Several modifications of ViT have been proposed to improve supervised learning, unsupervised learning, multi-scale processing, and dense predictions. In particular, DINO (Caron et al., 2021) uses a ViT within a self-supervised learning framework that performs self-distillation with exponential moving average updates. Caron et al. (2021) show that DINO's class-attention can produce localized and semantically meaningful salient object segmentations. Our work shows that DINO's features not only detect salient objects but can be used to extract dense and semantically meaningful correspondences between images. In STEGO, we refine the features of this pre-trained backbone to yield semantic segmentation predictions when clustered. We focus on DINO's embeddings because of their quality but note that STEGO can work with any deep network features.

## 3   METHODS

### 3.1   FEATURE CORRESPONDENCES PREDICT CLASS CO-OCCURRENCE

Recent progress in self-supervised visual feature learning has yielded methods with powerful and semantically relevant features that improve a variety of downstream tasks. Though most works aim to generate a single vector for an image, many works show that intermediate dense features are semantically relevant (Hamilton et al., 2021; Collins et al., 2018; Zhou et al., 2016). To use this information, we focus on the "correlation volume" (Teed & Deng, 2020) between the dense feature maps. For convolutional or transformer architectures, these dense feature maps can be the activation map of a specific layer. Additionally, the Q, K or V matrices in transformers can also serve as candidate features, though we find these attention tensors do not perform as well in practice. More formally, let $f \in \mathbb{R}^{CHW}, g \in \mathbb{R}^{CIJ}$ be the feature tensors for two different images where $C$ represents the channel dimension and $(H, W), (I, J)$ represent spatial dimensions. We form the feature correspondence tensor:

$$F_{hwij} := \sum_c \frac{f_{chw}}{|f_{hw}|} \frac{g_{cij}}{|g_{ij}|}, \qquad (1)$$

whose entries represent the cosine similarity between the feature at spatial position $(h, w)$ of feature tensor $f$ and position $(i, j)$ of feature tensor $g$. In the special case where $f = g$ these correspon-

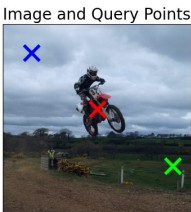 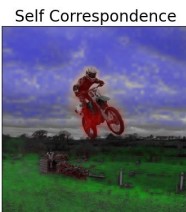 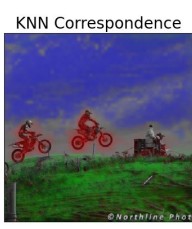
Image and Query Points | Self Correspondence | KNN Correspondence

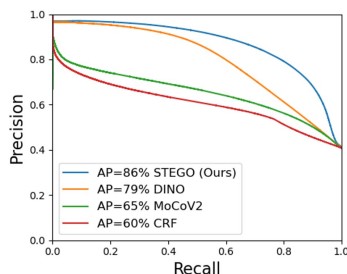

Figure 2: Feature correspondences from DINO. Correspondences between the source image (left) and the target images (middle and right) are plotted over the target images in the respective color of the source point (crosses in the left image). Feature correspondences can highlight key aspects of shared semantics within a single image (middle) and across similar images such as KNNs (right)

Figure 3: Precision recall curves show that feature self-correspondences strongly predict true label co-occurrence. DINO outperforms MoCoV2 and a CRF kernel, which shows its power as an unsupervised learning signal.

dences measure the similarity between two regions of the same image. We note that this quantity appears often as the "cost-volume" within the optical flow literature, and Hamilton et al. (2021) show this acts a higher-order generalization of Class Activation Maps (Zhou et al., 2016) for contrastive architectures and visual search engines. By examining slices of the correspondence tensor, $F$, at a given $(h, w)$ we are able to visualize how two images relate according the featurizer. For example, Figure 2 shows how three different points from the source image (shown in blue, red, and green) are in correspondence with relevant semantic areas within the image and its K-nearest neighbors with respect to the DINO (Caron et al., 2021) as the feature extractor.

This feature correspondence tensor not only allows us to visualize image correspondences but is strongly correlated with the true label co-occurrence tensor. In particular, we can form the ground truth label co-occurrence tensor given a pair of ground-truth semantic segmentation labels $k \in \mathcal{C}^{HW}, l \in \mathcal{C}^{IJ}$ where $\mathcal{C}$ represents the set of possible classes:

$$L_{hwij} := \begin{cases} 1, & \text{if } l_{hw} = k_{ij} \\ 0, & \text{if } l_{hw} \neq k_{ij} \end{cases}$$

By examining how well the feature correspondences, $F$, predict the ground-truth label co-occurrences, $L$, we can measure how compatible the features are with the semantic segmentation labels. More specifically we treat the feature correspondences as a probability logit and compute the average precision when used as a classifier for $L$. This approach not only acts as a quick diagnostic tool to determine the efficacy of features, but also allows us to compare with other forms of supervision such as the fully connected Conditional Random Field (CRF) (Krähenbühl & Koltun, 2011), which uses correspondences between pixels to refine low-resolution label predictions. In Figure 3 we plot precision-recall curves for the DINO backbone, the MoCoV2 backbone, the CRF Kernel, and our trained STEGO architecture. Interestingly, we find that DINO is already a spectacular predictor of label co-occurrence within the Coco stuff dataset despite **never seeing the labels**. In particular, DINO recalls $50\%$ of true label co-occurrences with a precision of $90\%$ and significantly outperforms both MoCoV2 feature correspondences and the CRF kernel. One curious note is that our final trained model is a better label predictor than the supervisory signal it learns from. We attribute this to the distillation process discussed in Section 3.2 which amplifies this supervisory signal and drives consistency across the entire dataset. Finally, we stress that our comparison to ground truth labels within this section is solely to provide intuition about the quality of feature correspondences as a supervisory signal. **We do not use the ground truth labels to tune any parameters of STEGO.**

## 3.2 DISTILLING FEATURE CORRESPONDENCES

In Section 3.1 we have shown that feature correspondences have the potential to be a quality learning signal for unsupervised segmentation. In this section we explore how to harness this signal to create pixel-wise embeddings that, when clustered, yield a quality semantic segmentation. In particular, we seek to learn a low-dimensional embedding that "distills" the feature correspondences. To achieve

this aim, we draw inspiration from the CRF which uses an undirected graphical model to refine noisy or low-resolution class predictions by aligning them with edges and color-correlated regions in the original image.

More formally, let $\mathcal{N} : \mathbb{R}^{C'H'W'} \to \mathbb{R}^{CHW}$ represent a deep network backbone, which maps an image $x$ with $C'$ channels and spatial dimensions $(H', W')$ to a feature tensor $f$ with $C$ channels and spatial dimensions $(H, W)$. In this work, we keep this backbone network frozen and focus on training a light-weight segmentation head $\mathcal{S} : \mathbb{R}^{CHW} \to \mathbb{R}^{KHW}$, that maps our feature space to a code space of dimension $K$, where $K < C$. The goal of $\mathcal{S}$ is to learn a nonlinear projection, $\mathcal{S}(f) =: s \in \mathbb{R}^{KHW}$, that forms compact clusters and amplifies the correlation patterns of $f$.

To build our loss function let $f$ and $g$ be two feature tensors from a pair of images $x$, and $y$ and let $s := \mathcal{S}(f) \in \mathbb{R}^{CHW}$ and $t := \mathcal{S}(g) \in \mathbb{R}^{CIJ}$ be their respective segmentation features. Next, using Equation 1 we compute a feature correlation tensor $F \in R^{HWIJ}$ from $f$ and $g$ and a segmentation correlation tensor $S \in R^{HWIJ}$ from $s$ and $t$. Our loss function aims to push the entries of $s$ and $t$ together if there is a significant coupling between two corresponding entries of $f$ and $g$. As shown in Figure 4, we can achieve this with a simple element-wise multiplication of the tensors $F$ and $S$:

$$\mathcal{L}_{simple-corr}(x, y, b) := -\sum_{hwij}(F_{hwij} - b)S_{hwij} \tag{2}$$

Where $b$ is a hyper-parameter which adds uniform "negative pressure" to the equation to prevent collapse. Minimizing $\mathcal{L}$ with respect to $S$ encourages elements of $S$ to be large when elements of $F - b$ are positive and small when elements of $F - b$ are negative. More explicitly, because the elements of $F$ and $S$ are cosine similarities, this exerts an attractive or repulsive force on pairs of segmentation features with strength proportional to their feature correspondences. We note that the elements of $S$ are not just encouraged to *equal* the elements of $F$ but rather to push to total anti-alignment $(-1)$ or alignment $(1)$ depending on the sign of $F - b$.

In practice, we found that $\mathcal{L}_{simple-corr}$ is sometimes unstable and does not provide enough learning signal to drive the optimization. Empirically, we found that optimizing the segmentation features towards total anti-alignment when the corresponding features do not correlate leads to instability, likely because this increases co-linearity. Therefore, we optimize weakly-correlated segmentation features to be orthogonal instead. This can be efficiently achieved by clamping the segmentation correspondence, $S$, at 0, which dramatically improved the optimization stability.

Additionally, we encountered challenges when balancing the learning signal for small objects which have concentrated correlation patterns. In these cases, $F_{hwij} - b$ is negative in most locations, and the loss drives the features to diverge instead of aggregate. To make the optimization more balanced, we introduce a **S**patial **C**entering operation on the feature correspondences:

$$F^{SC}_{hwij} := F_{hwij} - \frac{1}{IJ}\sum_{i'j'}F_{hwi'j'}. \tag{3}$$

Together with the zero clamping, our final correlation loss is defined as:

$$\mathcal{L}_{corr}(x, y, b) := -\sum_{hwij}(F^{SC}_{hwij} - b)max(S_{hwij}, 0). \tag{4}$$

We demonstrate the positive effect of both the aforementioned "0-Clamp" and "SC" modifications in the ablation study of Table 2.

### 3.3 STEGO ARCHITECTURE

STEGO uses three instantiations of the correspondence loss of Equation 4 to train a segmentation head to distill feature relationships between an image and itself, its K-Nearest Neighbors (KNNs), and random other images. The self and KNN correspondence losses primarily provide positive, attractive, signal and random image pairs tend to provide negative, repulsive, signal. We illustrate this and other major architecture components of STEGO in Figure 4.

STEGO is made up of a frozen backbone that serves as a source of learning feedback, and as an input to the segmentation head for predicting distilled features. This segmentation head is a simple

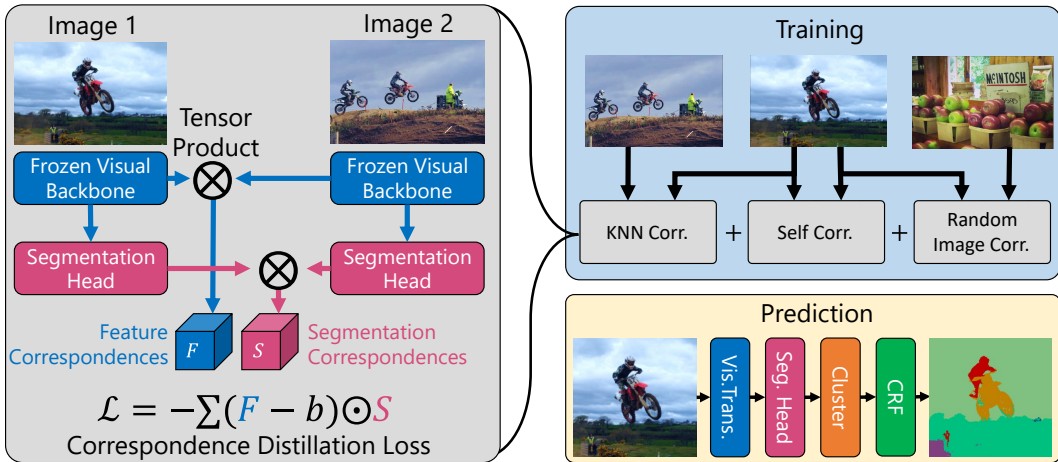

Figure 4: High-level overview of the STEGO architecture at train and prediction steps. Grey boxes represent three different instantiations of the main correspondence distillation loss which is used to train the segmentation head.

feed forward network with ReLU activations (Glorot et al., 2011). In contrast to other works, our method does not re-train or fine-tune the backbone. This makes our method very efficient to train: it only takes less than 2 hours on a single NVIDIA V100 GPU card.

We first use our backbone to extract global image features by global average pooling (GAP) our spatial features: $GAP(f)$. We then construct a lookup table of each image's K-Nearest Neighbors according to cosine similarity in the backbone's feature space. Each training minibatch consists of a collection of random images $x$ and random nearest neighbors $x^{knn}$. In our experiments we sample $x^{knn}$ randomly from each image's top 7 KNNs. We also sample random images, $x^{rand}$, by shuffling $x$ and ensuring that no image matched with itself. STEGO's full loss is:

$$\mathcal{L} = \lambda_{self}\mathcal{L}_{corr}(x, x, b_{self}) + \lambda_{knn}\mathcal{L}_{corr}(x, x^{knn}, b_{knn}) + \lambda_{rand}\mathcal{L}_{corr}(x, x^{rand}, b_{rand}) \quad (5)$$

Where the $\lambda$'s and the $b$'s control the balance of the learning signals and the ratio of positive to negative pressure respectively. In practice, we found that a ratio of $\lambda_{self} \approx \lambda_{rand} \approx 2\lambda_{knn}$ worked well. The $b$ parameters tended to be dataset and network specific, but we aimed to keep the system in a rough balance between positive and negative forces. More specifically we tuned the $b$s to keep mean KNN feature similarity at $\approx 0.3$ and mean random similarity at $\approx 0.0$.

Many images within the CocoStuff and Cityscapes datasets are cluttered with small objects that are hard to resolve at a feature resolution of $(40, 40)$. To better handle small objects and maintain fast training times we five-crop training images prior to learning KNNs. This not only allows the network to look at closer details of the images, but also improves the quality of the KNNs. More specifically, global image embeddings are computed for each crop. This allows the network to resolve finer details and yields five times as many images to find close matching KNNs from. Five-cropping improved both our Cityscapes results and CocoStuff segmentations, and we detail this in Table 2.

The final components of our architecture are the clustering and CRF refinement step. Due to the feature distillation process, STEGO's segmentation features tend to form clear clusters. We apply a cosine distance based minibatch K-Means algorithm (MacQueen et al., 1967) to extract these clusters and compute concrete class assignments from STEGO's continuous features. After clustering, we refine these labels with a CRF to improve their spatial resolution further.

### 3.4 RELATION TO POTTS MODELS AND ENERGY-BASED GRAPH OPTIMIZATION

Equation 4 can be viewed in the context of Potts models or continuous Ising models from statistical physics (Potts, 1952; Baker Jr & Kincaid, 1979). We briefly overview this connection, and point interested readers to Section A.8 for a more detailed discussion. To build the general Ising model, let $\mathcal{G} = (\mathcal{V}, w)$ be a fully connected, weighted, and undirected graph on $|\mathcal{V}|$ vertices. In our applications we take $\mathcal{V}$ to be the set of pixels in the training dataset. Let $w : \mathcal{V} \times \mathcal{V} \rightarrow \mathbb{R}$ represent an edge

Table 1: Comparison of unsupervised segmentation architectures on 27 class CocoStuff validation set. STEGO significantly outperforms prior art in both unsupervised clustering and linear-probe style metrics.

| Model | Unsupervised | | Linear Probe | |
|---|---|---|---|---|
| | Accuracy | mIoU | Accuracy | mIoU |
| ResNet50 (He et al., 2016) | 24.6 | 8.9 | 41.3 | 10.2 |
| MoCoV2 (Chen et al., 2020c) | 25.2 | 10.4 | 44.4 | 13.2 |
| DINO (Caron et al., 2021) | 30.5 | 9.6 | 66.8 | 29.4 |
| Deep Cluster (Caron et al., 2018) | 19.9 | - | - | - |
| SIFT (Lowe, 1999) | 20.2 | - | - | - |
| Doersch et al. (2015) | 23.1 | - | - | - |
| Isola et al. (2015) | 24.3 | - | - | - |
| AC (Ouali et al., 2020) | 30.8 | - | - | - |
| InMARS (Mirsadeghi et al., 2021) | 31.0 | - | - | - |
| IIC (Ji et al., 2019) | 21.8 | 6.7 | 44.5 | 8.4 |
| MDC (Cho et al., 2021) | 32.2 | 9.8 | 48.6 | 13.3 |
| PiCIE (Cho et al., 2021) | 48.1 | 13.8 | 54.2 | 13.9 |
| PiCIE + H (Cho et al., 2021) | 50.0 | 14.4 | 54.8 | 14.8 |
| **STEGO (Ours)** | **56.9** | **28.2** | **76.1** | **41.0** |

weighting function. Let $\phi : \mathcal{V} \to \mathcal{C}$ be a vertex valued function mapping into a generic code space $\mathcal{C}$ such as the probability simplex over cluster labels $\mathcal{P}(L)$, or the $K$-dimensional continuous feature space $\mathbb{R}^K$. The function $\phi$ can be a parameterized neural network, or a simple lookup table that assigns a code to each graph node. Finally, we define a compatibility function $\mu : \mathcal{C} \times \mathcal{C} \to \mathbb{R}$ that measures the cost of comparing two codes. We can now define the following graph energy functional:

$$E(\phi) := \sum_{v_i, v_j \in \mathcal{V}} w(v_i, v_j) \mu(\phi(v_i), \phi(v_j)) \tag{6}$$

Constructing the Boltzmann Distribution (Hinton, 2002) yields a normalized distribution over the function space $\Phi$:

$$p(\phi|w, \mu) = \frac{\exp(-E(\phi))}{\int_\Phi \exp(-E(\phi'))d\phi'} \tag{7}$$

In general, sampling from this probability distribution is difficult because of the often-intractable normalization factor. However, it is easier to compute the maximum likelihood estimate (MLE), $\arg\max_{\phi \in \Phi} p(\phi|w, \mu)$. In particular, if $\Phi$ is a smoothly parameterized space of functions and $\phi$ and $\mu$ are differentiable functions, one can compute the MLE using stochastic gradient descent (SGD) with highly-optimized automatic differentiation frameworks (Paszke et al., 2019; Abadi et al., 2015). In Section A.8 of the supplement we prove that the finding the MLE of Equation 7 is equivalent to minimizing the loss of Equation 4 when $|V|$ is the set of pixels in our image training set, $\phi = \mathcal{S} \circ \mathcal{N}$, $w$ is the cosine distance between features, and $\mu$ is cosine distance. Like STEGO, the CRF is also a Potts model, and we use this connection to re-purpose the STEGO loss function to create continuous, minibatch, and unsupervised variants of the CRF. We detail this exploration in Section A.9 of the Supplement.

## 4 EXPERIMENTS

We evaluate STEGO on standard semantic segmentation datasets and compare with current state-of-the-art. We then justify different design choices of STEGO through ablation studies. Additional details on datasets, model hyperparameters, hardware, and other implementation details can be found in Section A.10 of the Supplement.

### 4.1 EVALUATION DETAILS

**Datasets** Following Cho et al. (2021), we evaluate STEGO on the 27 mid-level classes of the CocoStuff class hierarchy and on the 27 classes of Cityscapes. Like prior art, we first resize images to 320 pixels along the minor axis followed by a $(320 \times 320)$ center crops of each validation image. We use mean intersection over union (mIoU) and Accuracy for evaluation metrics. Our CocoStuff evaluation setting originated in Ji et al. (2019) and is common in the literature. Our Cityscapes

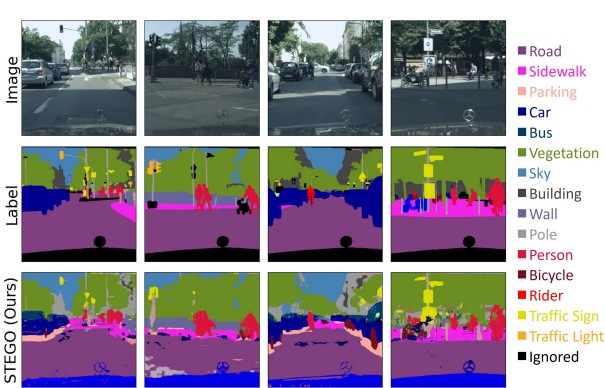

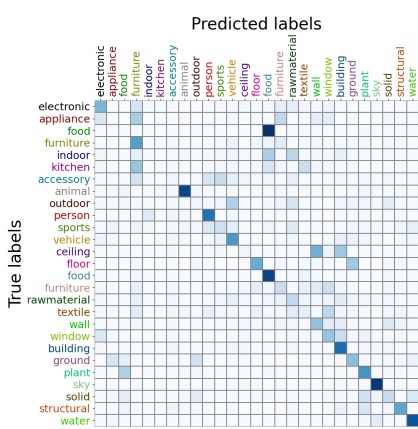

Figure 5: Comparison of ground truth labels (middle row) and cluster probe predictions for STEGO (bottom row) for images from the Cityscapes dataset.

Figure 6: Confusion matrix of STEGO cluster probe predictions on CocoStuff. Classes after the "vehicle" class are "stuff" and classes before are "things". Rows are normalized to sum to 1.

evaluation setting is adopted from Cho et al. (2021). The latter is newer and more challenging, and thus fewer baselines are available. Finally we also compare on the Potsdam-3 setting fro Ji et al. (2019) in Section A.2 of the Appendix.

**Linear Probe** The first way we evaluate the quality of the distilled segmentation features is through transfer learning effectiveness. As in Van Gansbeke et al. (2021); Cho et al. (2021); Chen et al. (2020b), we train a linear projection from segmentation features to class labels using the cross entropy loss. This loss solely evaluates feature quality and is not part of the STEGO training process.

**Clustering** Unlike the linear probe, the clustering step does not have access to ground truth supervised labels. As in prior art, we use a Hungarian matching algorithm to align our unlabeled clusters and the ground truth labels for evaluation and visualization purposes. This measures how consistent the predicted semantic segments are with the ground truth labels and is invariant to permutations of the predicted class labels.

## 4.2 RESULTS

We summarize our main results on the 27 classes of CocoStuff in Table 1. STEGO significantly outperforms the prior state of the art, PiCIE, on both linear probe and clustering (Unsupervised) metrics. In particular, STEGO improves by **+14** unsupervised mIoU, **+6.9** unsupervised accuracy, **+26** linear probe mIoU, and **+21** linear probe accuracy compared to the next best baseline. In Table 3, we find a similarly large improvement of **+8.7** unsupervised mIoU and **+7.7** unsupervised accuracy on the Cityscapes validation set. These two experiments demonstrate that even though we do not fine-tune the backbone for these datasets, DINO's self-supervised weights on ImageNet (Deng et al., 2009) are enough to simultaneously solve both settings. STEGO also outperforms simply clustering the features from unmodified DINO, MoCoV2, and ImageNet supervised ResNet50 backbones. This demonstrates the benefits of training a segmentation head to distill feature correspondences.

We show some example segmentations from STEGO and our baseline PiCIE on the CocoStuff dataset in Figure 1. We include additional examples and failure cases in Sections A.4 and A.5. We note that STEGO is significantly better at resolving fine-grained details within the images such as the legs of horses in the third image from the left column of Figure 1, and the individual birds in the right-most column. Though the PiCIE baseline uses a feature pyramid network to output high resolution predictions, the network does not attune to fine grained details, potentially demonstrating the limitations of the sparse training signal induced by data augmentations alone. In contrast, STEGO's predictions capture small objects and fine details. In part, this can be attributed to DINO backbone's higher resolution features, the 5-crop training described in 3.3, and the CRF post-processing which helps to align the predictions to image edges. We show qualitative results on the Cityscapes dataset in Figure 5. STEGO successfully identifies people, street, sidewalk, cars, and street signs with high

Table 2: Architecture ablation study on the CocoStuff Dataset (27 Classes).

| Arch. | 0-Clamp | 5-Crop | SC | CRF | Unsup. Acc. | Unsup. mIoU | Linear Probe Acc. | Linear Probe mIoU |
|---|---|---|---|---|---|---|---|---|
| MoCoV2 | ✓ | | | | 48.4 | 20.8 | 70.7 | 26.5 |
| ViT-S | | | | | 34.2 | 7.3 | 54.9 | 15.6 |
| ViT-S | ✓ | | | | 44.3 | 21.3 | 70.9 | 36.8 |
| ViT-S | ✓ | ✓ | | | 47.6 | 23.4 | 72.2 | 36.8 |
| ViT-S | ✓ | ✓ | ✓ | | 47.7 | 24.0 | 72.9 | 38.4 |
| ViT-S | ✓ | ✓ | ✓ | ✓ | 48.3 | 24.5 | 74.4 | 38.3 |
| ViT-B | ✓ | ✓ | ✓ | | 54.8 | 26.8 | 74.3 | 39.5 |
| ViT-B | ✓ | ✓ | ✓ | ✓ | **56.9** | **28.2** | **76.1** | **41.0** |

Table 3: Results on the Cityscapes Dataset (27 Classes). STEGO improves significantly over all baselines in both accuracy and mIoU.

| Model | Unsup. Acc. | Unsup. mIoU |
|---|---|---|
| IIC (Ji et al., 2019) | 47.9 | 6.4 |
| MDC (Cho et al., 2021) | 40.7 | 7.1 |
| PiCIE (Cho et al., 2021) | 65.5 | 12.3 |
| **STEGO (Ours)** | **73.2** | **21.0** |

detail and fidelity. We note that prior works did not publish pretrained models or linear probe results on Cityscapes so we exclude this information from Table 3 and Figure 5.

To better understand the predictions and failures of STEGO, we include confusion matrices for CocoStuff (Figure 6) and Cityscapes (Figure 11 of the Supplement). Some salient STEGO errors include confusing the "food" category from the CocoStuff "things", and the "food" category from CocoStuff "stuff". STEGO also does not properly separate "ceilings" from "walls", and lacks consistent segmentations for classes such as "indoor", "accessory", "rawmaterial" and "textile". These errors also draw our attention to the challenges of evaluating unsupervised segmentation methods: *label ontologies can be arbitrary*. In these circumstances the divisions between classes are not well defined and it is hard to imagine a system that can segment the results consistently without additional information. In these regimes, the linear probe provides a more important barometer for quality because the limited supervision can help disambiguate these cases. Nevertheless, we feel that there is still considerable progress to be made on the purely unsupervised benchmark, and that even with the improvements of STEGO there is still a measurable performance gap with supervised systems.

### 4.3 ABLATION STUDY

To understand the impact of STEGO's architectural components we perform an ablation analysis on the CocoStuff dataset, and report the results in Table 2. We examine the effect of using several different backbones in STEGO including MoCoV2, the ViT-Small, and ViT-Base architectures of DINO. We find that ViT-Base is the best feature extractor of the group and leads by a significant margin both in terms of accuracy and mIoU. We also evaluate the several loss function and architecture decisions described in Section 3.3. In particular, we explore clamping the segmentation feature correspondence tensor at 0 to prevent the negative pressure from introducing co-linearity (0-Clamp), five-cropping the dataset prior to mining KNNs to improve the resolution of the learning signal (5-Crop), spatially centering the feature correspondence tensor to improve resolution of small objects (SC), and Conditional Random Field post-processing to refine predictions (CRF). We find that these modifications improve both the cluster and linear probe evaluation metrics.

### 5 CONCLUSION

We have found that modern self-supervised visual backbones can be refined to yield state of the art unsupervised semantic segmentation methods. We have motivated this architecture by showing that correspondences between deep features are directly correlated with ground truth label co-occurrence. We take advantage of this strong, yet entirely unsupervised, learning signal by introducing a novel contrastive loss that "distills" the correspondences between features. Our system, STEGO, produces low rank representations that cluster into accurate semantic segmentation predictions. We connect STEGO's loss to CRF inference by showing it is equivalent to MLE in Potts models over the entire collection of pixels in our dataset. We show STEGO yields a significant improvement over the prior state of the art, on both the CocoStuff (**+14 mIoU**) and Cityscapes (**+9 mIoU**) semantic segmentation challenges. Finally, we justify the architectural decisions of STEGO with an ablation study on the CocoStuff dataset.

ACKNOWLEDGMENTS

We would like to thank Karen Hamilton for proofreading the work and Siddhartha Sen for sponsoring access to the Microsoft Research compute infrastructure. We also thank Jang Hyun Cho for helping us run and evaluate the PiCIE baseline. We thank Kavital Bala, Vincent Sitzmann, Marc Bosch, Desalegn Delelegn, Cody Champion, and Markus Weimer for their helpful commentary on the work.

This material is based upon work supported by the National Science Foundation Graduate Research Fellowship under Grant No. 2021323067. Any opinion, findings, and conclusions or recommendations expressed in this material are those of the authors(s) and do not necessarily reflect the views of the National Science Foundation. This research is based upon work supported in part by the Office of the Director of National Intelligence (Intelligence Advanced Research Projects Activity) via 2021-20111000006. The views and conclusions contained herein are those of the authors and should not be interpreted as necessarily representing the official policies, either expressed or implied, of ODNI, IARPA, or the U S Government. The US Government is authorized to reproduce and distribute reprints for governmental purposes notwithstanding any copyright annotation therein. This work is supported by the National Science Foundation under Cooperative Agreement PHY-2019786 (The NSF AI Institute for Artificial Intelligence and Fundamental Interactions, http://iaifi.org/)

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

# A    APPENDIX

## A.1    VIDEO AND CODE

We include a short video description of our work at https://aka.ms/stego-video.

We also provide training and evaluation code at https://aka.ms/stego-code

## A.2    ADDITIONAL RESULTS ON THE POTSDAM-3 DATASET

In addition to our evaluations in Section 4.1 we compare STEGO to prior art on the Potsdam 3-class aerial image segmentation task presented in Ji et al. (2019). In Table **??** We find that STEGO is able to achieve $+12\%$ accuracy compared to the previous state of the art, IIC. We show example qualitative results in Figure 7.

Table 4: Additional results on the Potsdam-3 aerial image segmentation challenge

| Model | Unsup. Acc. |
|---|---|
| Random CNN (Ji et al., 2019) | 38.2 |
| K-Means (Pedregosa et al., 2011) | 45.7 |
| SIFT (Lowe, 1999) | 38.2 |
| Doersch et al. (2015) | 49.6 |
| Isola et al. (2015) | 63.9 |
| Deep Cluster (Caron et al., 2018) | 41.7 |
| IIC (Ji et al., 2019) | 65.1 |
| **STEGO (Ours)** | **77.0** |

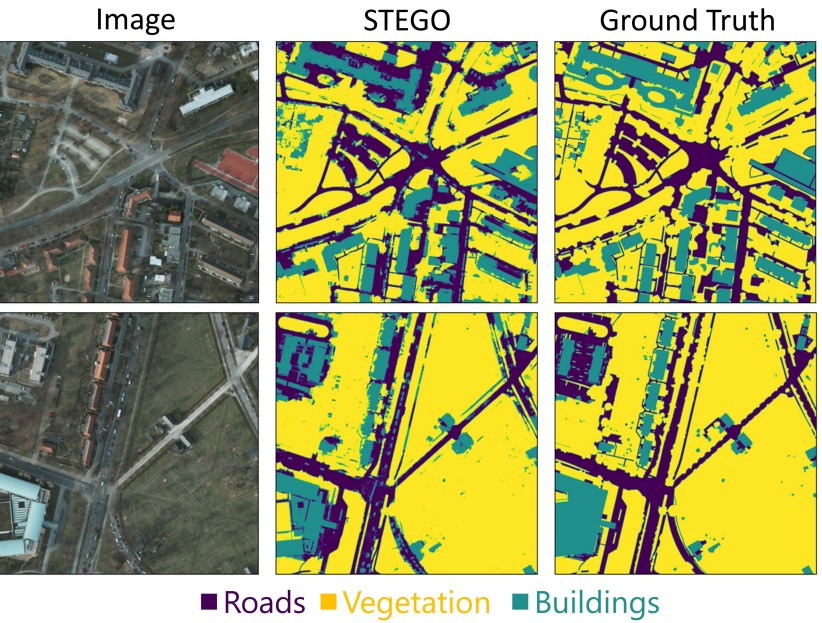

Figure 7: Qualitative comparison of STEGO segmentation results on the Potsdam-3 segmentation challenge.

## A.3 ADDITIONAL ABLATION STUDY

In addition to the ablation study of Table 2, we investigate the effect of each major architectural decision in isolation. We find that in most metrics, removing each architectural component hurts performance.

Table 5: Additional architecture ablation study on the CocoStuff Dataset (27 Classes).

| Backbone | 0-Clamp | 5-Crop | Pointwise | CRF | Self-Loss | KNN-Loss | Rand-Loss | Unsupervised Acc. | mIoU | Linear Probe Acc. | mIoU |
|---|---|---|---|---|---|---|---|---|---|---|---|
| ViT-Small | ✓ | ✓ | ✓ | ✓ | ✓ | ✓ | ✓ | **48.3** | **24.5** | **74.4** | 38.3 |
| MoCoV2 | ✓ | ✓ | ✓ | ✓ | ✓ | ✓ | ✓ | 43.1 | 19.6 | 65.9 | 26.0 |
| ViT-Small | | ✓ | ✓ | ✓ | ✓ | ✓ | ✓ | 42.8 | 10.3 | 59.3 | 19.3 |
| ViT-Small | ✓ | | ✓ | ✓ | ✓ | ✓ | ✓ | 48.0 | 23.1 | 73.9 | **38.9** |
| ViT-Small | ✓ | ✓ | | ✓ | ✓ | ✓ | ✓ | 50.2 | 22.3 | 73.7 | 37.7 |
| ViT-Small | ✓ | ✓ | ✓ | | ✓ | ✓ | ✓ | 47.7 | 24.0 | 72.9 | 38.4 |
| ViT-Small | ✓ | ✓ | ✓ | ✓ | | ✓ | ✓ | 43.0 | 20.2 | 73.0 | 36.2 |
| ViT-Small | ✓ | ✓ | ✓ | ✓ | ✓ | | ✓ | 47.0 | 22.2 | 74.0 | 37.7 |
| ViT-Small | ✓ | ✓ | ✓ | ✓ | ✓ | ✓ | | 39.8 | 12.8 | 65.5 | 29.9 |

A.4    ADDITIONAL QUALITATIVE RESULTS

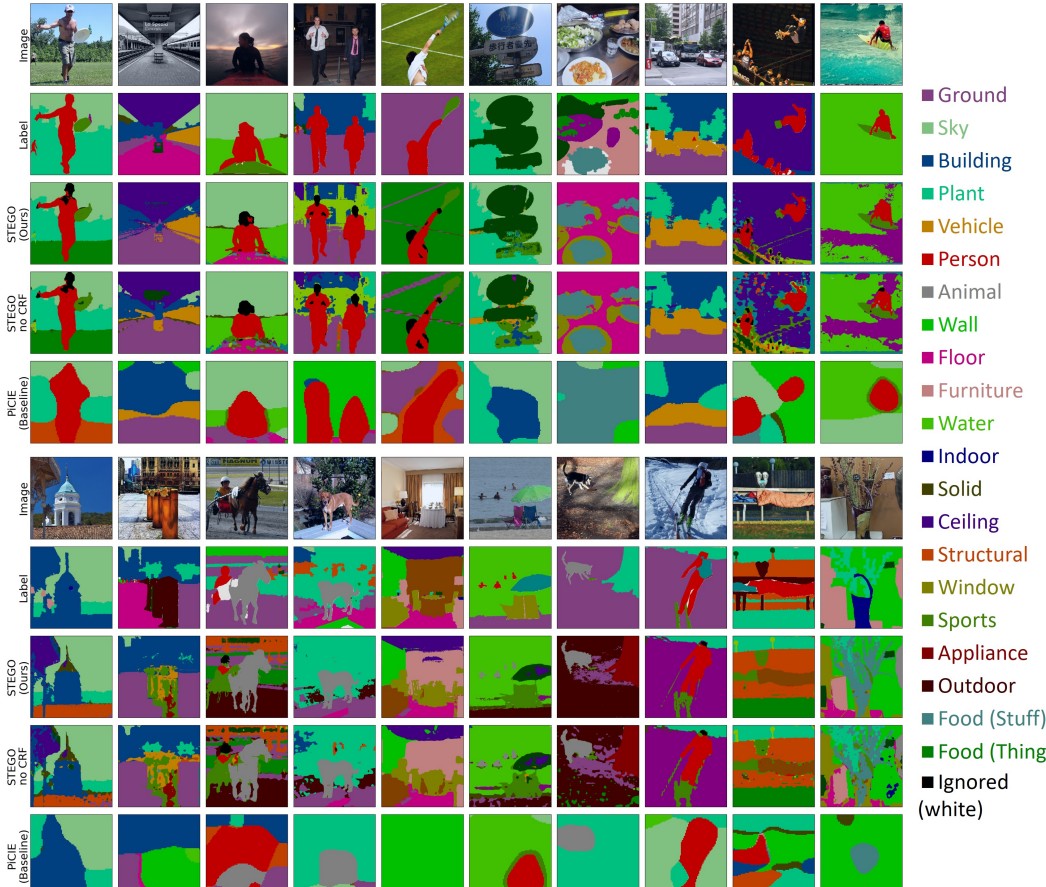

Figure 8: Additional unsupervised semantic segmentation predictions on the CocoStuff 27 class segmentation challenge using STEGO (Ours) and the prior state of the art, PiCIE. Images are not curated.

## A.5 FAILURE CASES

Unsupervised Segmentation is prone to a variety of issues. We include some of the following to segmentations to demonstrate cases where STEGO breaks down. In the first column of Figure 9 we can see that STEGO improperly segments ground from trees and backgrounds. In the second column we see that STEGO makes an understandable error and assigns the barn floor to the "outdoor" class and the barn wall to the "building" class. In the third column STEGO misses the boundary between wall and ceiling. The fourth column demonstrates the challenge between food (thing) and food (stuff) characterization. Interestingly PiCIE makes the same type of error both here, and in the barn case. The last column shows an example of STEGO missing a human in the lower left. In this image it is challenging to spot the person, probably because it is grayscale.

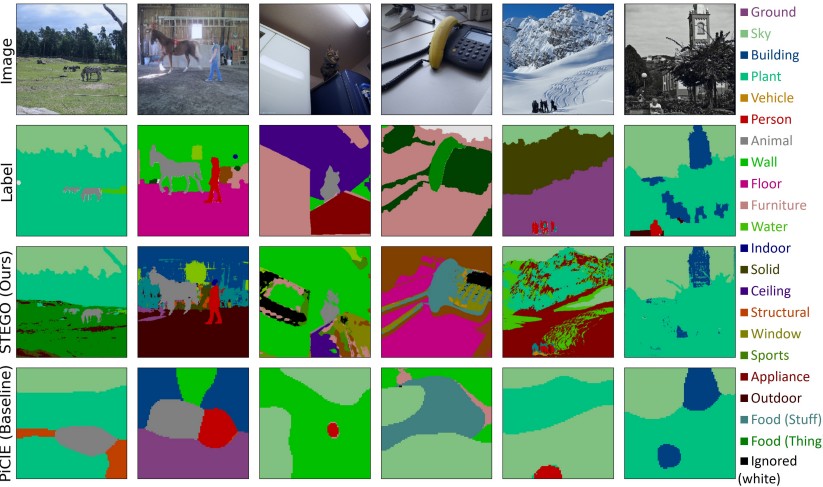

Figure 9: STEGO failure cases.

A.6   Feature Correspondences Predict STEGO's Errors

Section 3.1 demonstrates how unsupervised feature correspondences serve as an excellent proxy for the true label co-occurrence information. In this section we explore how and where DINO's feature correspondences systematically differ from the ground truth labels, and show that these insights allow us to directly predict STEGO's final confusion matrix.

More specifically we consider the setting of Section 3.1. Instead of computing precision-recall curves from our feature correspondence scores we can instead threshold these scores, select the strongest couplings between the images, and evaluate whether these couplings are between objects of the same class or objects of different classes. In particular, Figure 10 shows a confusion matrix capturing how well DINO feature correspondences between images and their K-Nearest Neighbors align with the ground truth label ontology in the CocoStuff27 dataset. We find that that this analysis predicts many of the areas where the final STEGO architecture fails. In particular, we can see that DINO conflates the "Food (things)" and "Food (stuff)" and this error also appears in STEGO's confusion matrix in Figure 12. Likewise both visualizations show confusion between "appliance" and "furniture", "window" and "wall", and several other common errors.

This analysis demonstrates that many of STEGO's errors originate from the structure of the DINO features used to train STEGO as opposed to other aspects of the architecture. However we note that the question of whether whether this is an issue with the DINO features, or due to ambiguities in the CocoStuff label ontology is still outstanding. Finally we note that this analysis is able to predict the results of a fully-trained STEGO architecture, and could be used as a way to select better backbones without having to training STEGO.

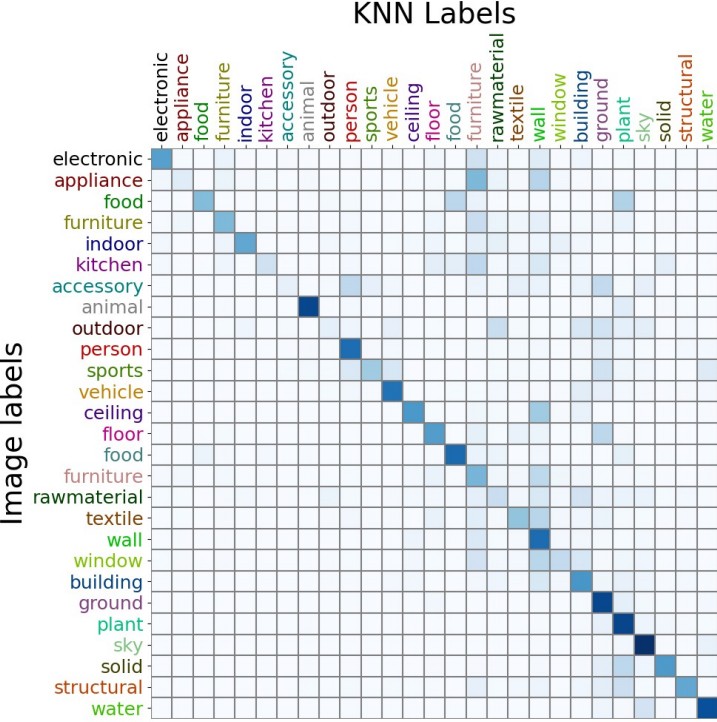

Figure 10: Normalized matrix of predicted label co-occurrences between an Images and KNNs. This analysis shows where our unsupervised supervisory signal, the DINO feature correspondences, fails to align with the CocoStuff27 label ontology.

## A.7 HIGHER RESOLUTION CONFUSION MATRICES

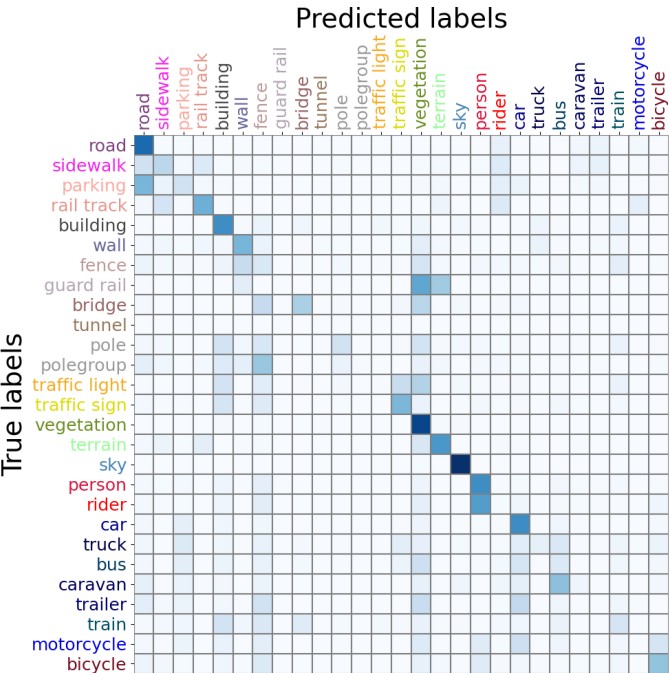

Figure 11: Confusion Matrix for Cityscapes predictions

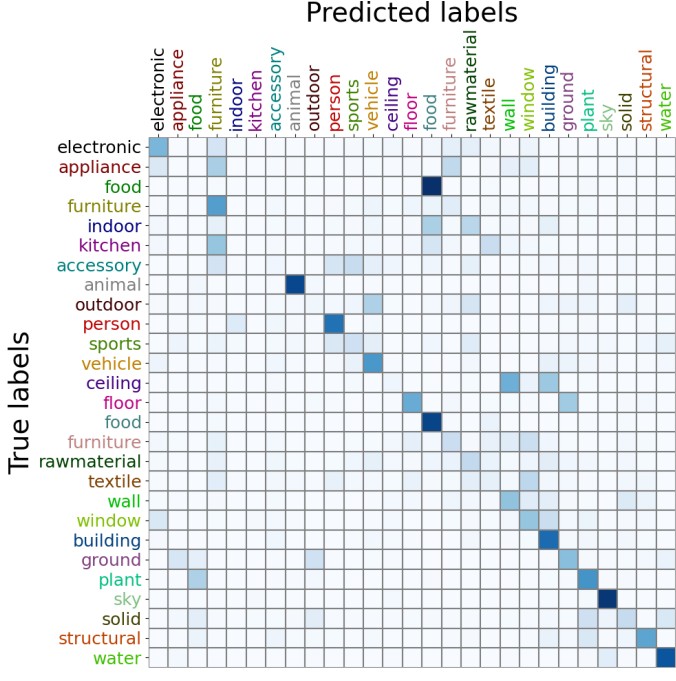

Figure 12: Confusion Matrix for CocoStuff predictions

A.8   Relationship with Graph Energy Minimization

In section 3.4 we briefly mention that STEGO's feature correlation distillation loss defined in Equation 4 can be seen as a particular case of Maximum Likelihood (ML) estimation on a undirected graphical model or Ising model. In this section we demonstrate this connection in greater detail using the formalism defined in 3.4. In particular, we recall the energy for a Potts model:

$$E(\phi) := \sum_{v_i, v_j \in \mathcal{V}} w(v_i, v_j) \mu(\phi(v_i), \phi(v_j)) \tag{8}$$

We then construct the Boltzmann Distribution (Hinton, 2002) yields a normalized distribution over the function space $\Phi$:

$$p(\phi | w, \mu) = \frac{\exp(-E(\phi))}{\int_\Phi \exp(-E(\phi'))d\phi'} \tag{9}$$

In general, sampling from this probability distribution is difficult because of the often-intractable normalization factor. However, it is easier to compute the maximum likelihood estimate (MLE):

$$\arg\max_{\phi \in \Phi} p(\phi | w, \mu) = \arg\max_{\phi \in \Phi} \frac{1}{Z} \exp(-E(\phi)) \tag{10}$$

Where $Z$ is the unknown constant normalization factor. Simplifying the right-hand side yields:

$$\arg\max_{\phi \in \Phi} p(\phi | w, \mu) = \arg\min_{\phi \in \Phi} E(\phi) = \arg\min_{\phi \in \Phi} \sum_{v_i, v_j \in \mathcal{V}} w(v_i, v_j) \mu(\phi(v_i), \phi(v_j)) \tag{11}$$

We are now in the position to connect this to the STEGO loss function. First, we take our nodes $\mathcal{V}$ to be the set of all spatial locations across our entire dataset of images. For concreteness we can represent $v \in \mathcal{V}$ by the tuple $(n, h, w)$ where $h, w$ represent height and width $n$ represents the image number. We now let $\phi(v_i)$ be the output of the segmentation head, $s_{v_i}$, at the image and spatial location $v_i$. Using cosine distance, $d_{cos}(x, y) = 1 - \frac{x}{|x|}\frac{y}{|y|}$ as the compatibility function, $\mu$, yields the following:

$$= \arg\min_{\mathcal{S}} \sum_{v_i, v_j \in \mathcal{V}} -w(v_i, v_j) \frac{s_{v_i}}{|s_{v_i}|} \frac{s_{v_j}}{|s_{v_j}|} \tag{12}$$

Wherte the argmin now ranges over the parameters of the segmentation head $\mathcal{S}$. We can now observe that the sum over all pairs $v_i, v_j \in \mathcal{V}$ can be written as a sum over pairs of images $x, y \in X$ and pairs of spatial locations $(h, w), (i, j)$ where we note that $(i, j)$ in this context refers to the spatial coordinates of image $y$ as in 3.1 and not the indices of the vertices.

$$= \arg\min_{\mathcal{S}} \sum_{x, y \in X} \sum_{hwij} -W(x, y)_{hwij} S(x, y)_{hwij} \tag{13}$$

Where we define $S(x, y)$ to be the segmentation feature correlation tensor for images $x, y$ as defined in Section 3.2. Finally letting $W(x, y)_{hwij} = F_{hwij} - b$ we recover our loss:

$$\arg\max_{\phi \in \Phi} p(\phi | w, \mu) = \arg\min_{\mathcal{S}} \sum_{x, y \in X} \mathcal{L}_{simple-corr}(x, y, b) \tag{14}$$

Finally we note that in practice we approximate the minimization using minibatch SGD, and our inclusion of KNN and Self-correspondence distillation changes the weight function $w$, but does not change its functional form.

Switching to the ML formulation of this problem allows us to solve this optimization for $\phi$ by gradient descent on the parameters of the segmentation head, $\mathcal{S}$, and makes this computationally tractable. For large image datasets that can contain millions of high-resolution images, the induced graph can contain billions of image locations. Other graph embedding and clustering approaches such as Spectral methods require solving for eigenvalues of the graph Laplacian, which can take $O(|\mathcal{V}|^3)$ time (Yan et al., 2009). More recent attempts to accelerate Spectral clustering such as (Yan et al., 2009) and (Han & Filippone, 2017) further assume a "Nonparametric" structure on the function $\phi$, where a separate cluster assignment is learned for each vertex. This assumption of a "nonparametric" function $\phi$ can be undesirable as one cannot cluster or embed new data without recomputing the entire clustering. In contrast, STEGO's backbone and segmentation head act as a parametric form for the function $\phi$ allowing the approach to output predictions for novel images.

### A.9 CONTINUOUS, UNSUPERVISED, AND MINI-BATCH CRF

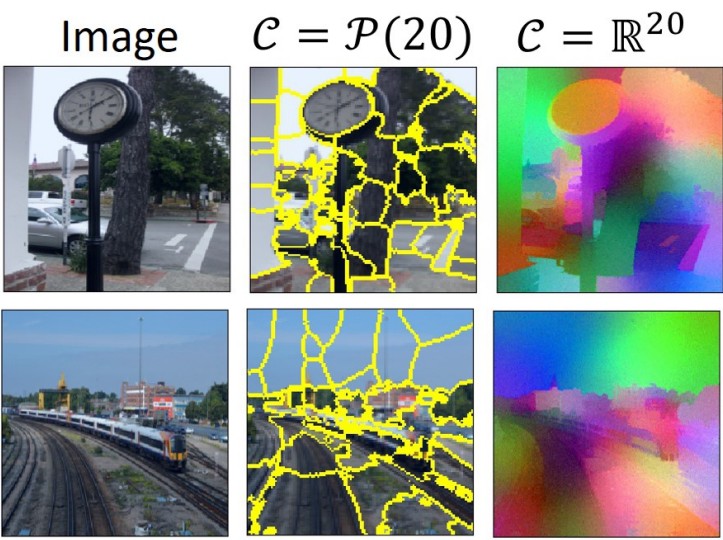

Figure 13: Unsupervised CRF solutions for discrete (middle) and continuous (right) code spaces. In the discrete case we mark the boundaries between classes, in the continuous case we visualize the top 3 dimensions of the code space.

Fully connected Gaussian Conditional Random Fields (CRFs) (Lafferty et al., 2001) are an extremely popular addition to semantic segmentation architectures. The CRF has the ability to improve initial predictions of locations, and can "sharpen" predictions to make them consistent with edges and areas with consistent color in the original image. CRF post-processing for refining supervised and weakly supervised semantic segmentation predictions is ubiquitous in the literature (Lafferty et al., 2001; Chen et al., 2014; Long et al., 2015; Liu et al., 2020; Ahn et al., 2019). Recently, new connections between CRF message passing and convolutional networks have allowed CRFs to be embedded into existing models (Chen et al., 2017; Teichmann & Cipolla, 2018) and trained jointly for better performance. By connecting the STEGO correspondence distillation loss to the energy of an undirected model on image pixels we can use the same minibatch MLE strategy to estimate other similar graphical models. For example, in the fully connected Gaussian edge potential CRF, one forms a pairwise potential function potential function for the pixels of a single image:

$$w_{crf}(v_i, v_j) = a \exp\left(-\frac{|p_i - p_j|^2}{2\theta_\alpha^2} - \frac{|I_i - I_j|^2}{2\theta_\beta^2}\right) + b \exp\left(-\frac{|p_i - p_j|^2}{2\theta_\gamma^2}\right) \quad (15)$$

Where $p_i$ represent the pixel coordinates associated with node $v_i$ and $I_i$ represents pixel colors associated with node $v_i$. The parameters $a, b, \theta_\alpha, \theta_\beta, \theta_\gamma$ are hyperparameters and control the behavior of the model. These parameters balance the effect of long- and short-range color similarities against

smoothness. The CRF directly learns a pixel-wise array of probabilistic class assignments over $k$ labels corresponding to the probability simplex code space $\mathcal{C} = \mathcal{P}(l)$ and a non-parametric clustering function $f$. For a compatibility function $\mu$ the CRF chooses the Potts Model (Potts, 1952): $\mu_{potts}(\phi(v_i), \phi(v_j)) := \mathbb{P}(\phi(v_i) \neq \phi(v_j))$.

With this setting of the weights and compatibility function, we directly recover the binary potentials of the fully connected Gaussian edge potential CRF (Krähenbühl & Koltun, 2011). We can also add the unary potentials which are often the outputs of another model. However, for our analysis we explore the case without unary potentials which yields an "unsupervised" variant of the CRF. However, without external unary potential terms, the strictly positive similarity kernel encourages the maximum likelihood estimator (MLE) of the graph to be the constant function. To rectify this, we can add small negative constant, $-b$, to the weight tensor to push unrelated pixels apart. This negative force is the direct analogue of the negative pressure hyper-parameter in STEGO and can be interpreted through the lens of negative sampling (Mikolov et al., 2013). This negative shift also appears in the word2vec and graph2vec embedding techniques (Narayanan et al., 2017; Levy & Goldberg, 2014). Our shifted CRF potential encourages natural clusters to form that respect the structure of the potentials that capture similarities in pixel colors and locations. In the discrete case, solutions to this equation resemble superpixel algorithms such as SLIC (Zhang et al., 2015). Additionally lifting this to the continuous code space and provide a natural continuous generalization of superpixels and seems to avoid challenging local minima. We illustrate these solutions to just the unsupervised CRF potential in Figure 13. Finally, we note that the second term of Equation 15, referred to as the smoothness kernel, matches IIC's notion of local class consistency. However, we found that adding these CRF terms to the self-correspondence loss of STEGO did not improve performance.

A.10   IMPLEMENTATION DETAILS

**Model**   STEGO uses the "ViT-Base" architecture of DINO pre-trained on ImageNet. This backbone was trained using self-supervision without access to ground-truth labels. We use the "teacher" weights when creating our backbone. We take the final layer of spatially varying features and apply a small amount ($p = 0.1$) of channel-wise dropout (Srivastava et al., 2014) before using them throughout the architecture during training. Our segmentation head consists of a linear network and a two-layer ReLU MLP added together and outputs a 70 dimensional vector. We use the Adam optimizer (Kingma & Ba, 2014) with a learning rate of 0.0005 and a batch size of 32. To make our losses resolution independent we sample 121 random spatial locations in the source and target implementations and use grid sampling (Jaderberg et al., 2015) to sample features from the backbone and segmentation heads. Our cluster probe is trained alongside the STEGO architecture using a minibatch k-means loss where closeness is measured by cosine distance. Cluster and linear probes are trained with separate Adam optimizers using a learning rate of .005

**Datasets**   We use the training and validation sets of Cocostuff described first in Ji et al. (2019) and used throughout the literature including in Cho et al. (2021). We note that the validation set used in Ji et al. (2019) is a subset of the full CocoStuff validation set and we use this validation subset to be consistent with prior benchmarks. We note that using the full validation set does not change results significantly. When five-cropping images we use a target size of $(.5h, .5w)$ for each crop where $h, w$ are the original image height and width. Training images are then scaled to have minor axis equal to 224 and are then center cropped to $(224, 224)$, validation images are first scaled to 320 then are center cropped to $(320, 320)$. All image resizing uses bilinear interpolation and resizing of target tensors for evaluation uses nearest neighbor interpolation.

**CRF**   We use PyDenseCRF (Krähenbühl & Koltun, 2011) with 10 iterations with parameters $a = 4, b = 3, \theta_\alpha = 67, \theta_\beta = 3, \theta_\gamma = 1$ as written in Section A.9.

**Compute**   All experiments use PyTorch (Paszke et al., 2019) v1.7 pre-trained models, on an Ubuntu 16.04 Azure NV24 Virtual Machine with Python 3.6. Experiments use PyTorch Lightning for distributed and multi-gpu training when necessary (Falcon et al., 2019).

**Hyperparameters**   We use the following hyperparameters for our results in Tables 1 and 3:

Table 6: Hyperparameters used in STEGO

| Parameter | Cityscapes | CocoStuff |
|:---:|:---:|:---:|
| $\lambda_{rand}$ | 0.91 | 0.15 |
| $\lambda_{knn}$ | 0.58 | 1.00 |
| $\lambda_{self}$ | 1.00 | 0.10 |
| $b_{rand}$ | 0.31 | 1.00 |
| $b_{knn}$ | 0.18 | 0.20 |
| $b_{self}$ | 0.46 | 0.12 |

## A.11 A HEURISTIC FOR SETTING HYPER-PARAMETERS

Setting hyperparameters without cross-validation on ground truth data can be difficult and this is an outstanding challenges with the STEGO architecture that we hope can be solved in future work. Nevertheless we have identified some key intuition to guide manual hyperparameter tuning. More specifically, we find that the most important factor affecting performance is the balance of positive and negative forces. Too much negative feedback and vectors will all push apart and clusters will not form well, too much positive feedback and the system will tend towards a small number of clusters. To debug this balance, we found it useful to visualize the distribution of feature correspondence similarities as a function of training step as shown in Figure 14. A balanced system (Orange distribution) will tend towards a bi-modal distribution with peaks at alignment 1 or orthogonality at 0. This bi-modal structure is indicative that there is some clustering within images, but that not everything is assigned to the same cluster. Pink and blue distributions show too much positive and negative signal respectively. We find that given a reasonable balance of the $\lambda$'s, this balance can be achieved by tuning the $b$s to achieve the desired balance.

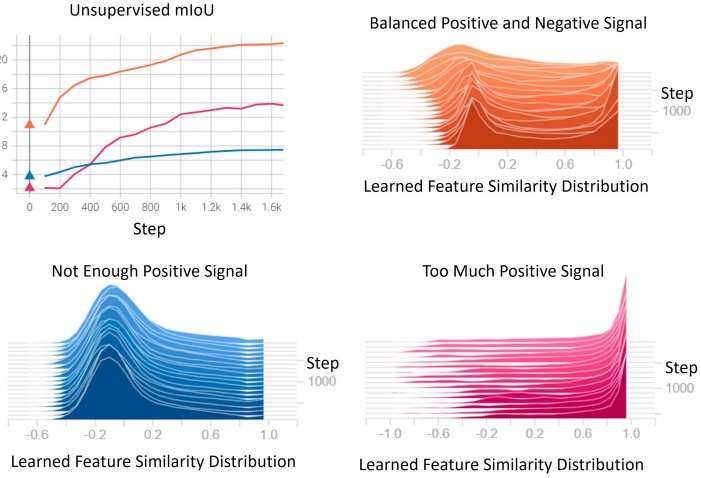

Figure 14: Distributions of feature correspondences between an image and itself across three different hyper-parameter settings. The orange curve and distribution shows a proper balance between attractive and repulsive forces allowing some pairs features to cluster together (the peak at 1) and other pairs of features to orthogonalize (the peak at 0)

### A.12 A NOTE ON 5-CROP NEAREST NEIGHBORS

We found that pre-processing the dataset by 5-cropping images was a simple and effective way to improve the spatial resolution of STEGO and the quality of K-Nearest Neighbors. We consider each resulting 5-crop as a separate image when computing KNNs and patches from the same image are valid KNNs. Figure 15 shows the distribution of these self-matches for the CocoStuff dataset. We note that the majority of patches do not have any nearest neighbors from the same image.

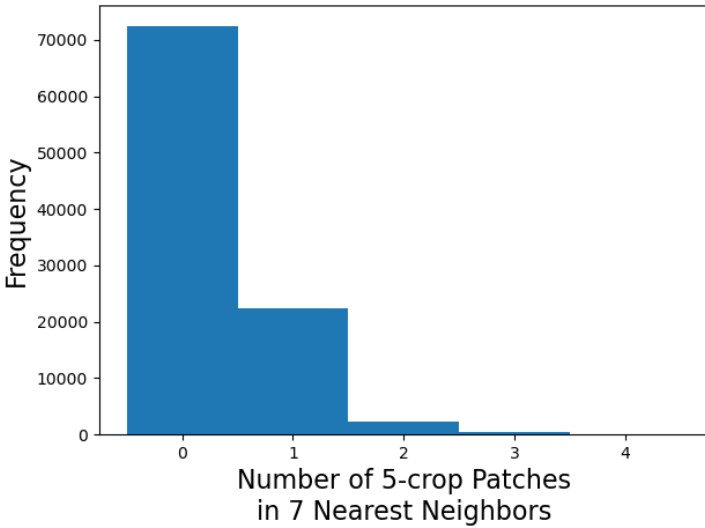

Figure 15: Number of patches from the same image found within each patch's 7 nearest neighbors

