# OpenReview forum: "Unsupervised Semantic Segmentation by Distilling Feature Correspondences"
_ICLR.cc/2022/Conference — ICLR 2022 Poster_

### Official Review · Reviewer_GKTw · 2021-11-02

**Correctness:** 3
**Technical Novelty And Significance:** 2
**Empirical Novelty And Significance:** 2
**Recommendation:** 6
**Confidence:** 2

**Main Review:**

Strength:
Overall, writing of this paper is clear and straightforward. STEGO achieves state of the art performance on both the CocoStuff and Cityscapes segmentation challenges.

Weakness:
1)	Novelty:
The main contribution is that it trained the additional segmentation head with the proposed contrastive loss over the frozen visual backbone. The novelty of utilizing the contrastive learning to form feature clusters is not so obvious.

2)	Implementation:
First, the approach seems complex to implement. For example, there are too many hyperparameters (eg, b and lamba) in the contrastive loss definition and it seems that the tunning of the parameters is complex.
Second, as is pointed out by the authors, the optimization is sometimes unstable. Adding clamping step seems not completely solves this problem.

3)	Experiment：
The state-of-the-art segmentation performance may be due to many factors such as strong DINO backbone features, CRF processing etc. Hoping more ablation studies are performed on each component of the proposed contrastive loss to verify the effect. For example, more ablation studies based on the ViT-Base architectures of DINO are preferred. The ablation studies on the hyperparameters are also preferred to be added.







**Summary Of The Paper:**

In this paper, the author proposes an unsupervised semantic segmentation approach STEGO. The core of STEGO is a novel contrastive loss function that encourages features to form compact semantic clusters.

**Summary Of The Review:**

I tend to weakly reject the paper based mainly based on the concerns about implementation and experiments mentioned above.

---

> ### Author Response · Authors · 2021-11-15
> **Responding to GKTw**
>
> We thank the reviewer for the helpful comments, and would like to clarify some of the concerns raised:
>
> ## Novelty
> We emphasize that one major contribution of our work is to propose an unsupervised semantic segmentation method that disentangles segmentation generation and feature extraction. The design of our method is motivated by this goal, and we have made a concrete effort that turns this motivation into a state-of-the-art unsupervised semantic segmentation method. In addition, through our experiments and our new additional ablation study, we justified the design decisions of our method. We believe a reductionism view is not a suitable way to evaluate the novelty of our work.
>
> ## Implementation
> We provided an additional analysis on choosing the hyperparameters b and lambda in our updated appendix A.10. The analysis showed that the distribution of the feature correlation is indicative of the model performance, where a bi-modal distribution is preferred. This process is not complex, but it does require tuning. We mention this as a limitation of our method and would like to address this issue in future work. Aside from the choice of b and lambda, we do not encounter any difficulties making the optimization successful. Finally, we note that many other methods in the literature such as our baseline PiCIE have a similar number of hyper-parameters weighting the effect of different learning signals and that this should not be a reason to reject the work.
>
> Finally, We would like to point out that selecting hyperparameters is different from making the optimization stable. 0-clamping is essential for our method, without which there’s no right combination of hyper-params that will render reasonable results. We kindly disagree with your statement that “Adding clamping step seems not completely solves this problem.” as it dramatically improved convergence and performance.
>
> ## Experiments
> We provided additional ablations on the choice of the backbone, the loss terms, and with/without CRF. Our method is still state-of-the-art using ResNet as the backbone, or without the CRF refinement. The ablation on the losses justified our design choices.
>
> Please refer to our updated manuscript and the general response for more details.
>
> Thank you again for helping us improve this work.

---

### Official Review · Reviewer_cF6T · 2021-11-02

**Correctness:** 3
**Technical Novelty And Significance:** 3
**Empirical Novelty And Significance:** 2
**Recommendation:** 8
**Confidence:** 4

**Main Review:**

Overall I like the idea of this paper and vote for acceptance. The idea of distilling frozen and general self-supervised features for segmentation specific features is neat. Though as expected, there are several regularisations and 'tricks' to really make the idea work in practice, they are well-motivated and valitdated in the ablation experiments.

My major concern is about more clarifications of details and some additional ablation models (see cons below). Hopefully the authors can address my concern in the rebuttal period.

Pros:
- The paper is well written and easy to follow. The key design choices in the paper are well motivated and described in the paper.
- The idea of pushing features from self-supervised representation learning towards segmentation task-specific ones is really interesting, simple and effective.
-Extensive experiments are done to validate the system and promising segmentation quality is achieved.

Cons/Suggestions:

(1) The discussion in Sec 3.3 could be better arranged in my opinion and it is easier for readers to follow the main idea by jumping from Sec 3.2 to Sec. 3.4. I agree the discussion is helpful to connect the idea of this method to undirected graphical model and can be put into later part of paper.

(2) Is the choice of distance very improtant generally? In addition to consine distance, do normalised l2-distance or RBF/Gaussian perform similarly? I think the choice of distance or similarity metric is quite important and worth more discussions.

(3) I would like to see more qualitative results of STEGO without CRF step as CRF usually smooths out the predictions. It is more informative for readers to know the "raw" segmentations before CRF to better understand the systems. Do segmentations before CRF usually become noisy or have coarse boundries aligned with images?

(4) More clarifications:
- Since 5 crops are used before KNN sampling, is there a tendency to select other crops from the same image instead of different images? If it is true, is it intended to do so ? More clarifications would be good.

- Similar to last point,  in Table 2, the effectiveness of 5-crop and SC are not properly adjustified. How about the performance with a full  STEGO model exluding 5-crop or SC only, like the ablation of CRF. I am wondering, instead of adding on top of vanilla baselines, if the contribution of these designs will be less prominent when other modules already exist?

- I was a bit concerned about the huge computational cost in constructing dense correspondence tensors F and S at the beginning. In supplement it is mentioend that 121 samples are taken per step per training image, is this used to alleviate the cost of building these dense matrix? Will STEGO work better given larger training images with more closer details?


Some typos:

(1) Sec 2 Unsupervised Semantic Segmentation: between an an-> between an.

(2)  Sec 4.2: We show some examples segmentations -> We show some example segmentations.


**Summary Of The Paper:**

This paper proposed STEGO, an unsupervised approach for semantic clustering/segmentation using feature refinement on top of self-supervised neural networks. A distilled version of self-supervised features is learned segmentation specifically through a feed-forward network via SGD.

The main contribution comes from an idea of distilling powerful deep features from strong self-supervised backbones to further improve their semantic discriminativeness. Though the idea is intuitive, several loss functions and regularisations are proposed to avoid trivial solutions and make the idea really work in practice. Extensive qualitative and quantitatve results are done to demonstrate the performance of STEGO and its design choices.

**Summary Of The Review:**

I have carefully read the paper and like the idea of this paper in making self-supervised representation leanring towards dense predicton tasks like segmentation/clustering without any explicit mannual supervision.

I am aware of pros and cons in this works and would like to vote for acceptance.

---

> ### Author Response · Authors · 2021-11-15
> **Responding to questions and comments**
>
> Thank you for your thoughtful and detailed comments on our work. We have provided a high-level overview of major additions in the general rebuttal and hope to address more targeted concerns in this comment.
>
> ## Moving discussion on Potts Models
> We have moved the section on Potts models to after our description of the architecture as requested.
>
> ## The choice of distance function
> In our work, we use cosine similarity between features as the “distance” function between features though in principle any distance can be used in the feature correlation tensor. We chose cosine similarity because
> it is common in the contrastive learning and ML literature (Appears in Moco, SimCLR, Word2Vec, etc) and has shown superior performance for retrieval in high dimensional embedding spaces
> Is bounded between -1 and 1 which allows us to get an intuitive sense of how aligned, orthogonal, or anti-aligned two vectors are
> Is invariant to scalar multiplication of the input features which helps us keep things like the learning rate consistent across different backbones
>
> ## Are 5 crops KNNs within the same image or across images?
>
> We added an extra section (A11) to address this question further. We found that the majority of images do not match with a patch from the same image in their top 7 KNNs. Furthermore, we clarify that KNNs between patches of the same image are part of the intended method as they help enforce consistency across image patches.
>
> ## Computational cost of dense correspondence (CF6T)
>
> You are correct that we use these 121 samples to approximate the signal from the full correspondence tensor. We highlight that our decision to randomly sample 121 feature locations in each image with grid sampling is already aimed to alleviate the computational burden of computing full feature correspondences without sacrificing accuracy or fine details. By sampling random locations on each training step, we stochastically approximate the signal that arises from the full 1600 (40x40) feature locations. We found this approach effectively decouples STEGO from the resolution of the backbone’s feature maps and could achieve the same end performance with a fraction of the memory and computation time. Moreover, our final results show resolution on the order of the full feature map as opposed to the sampling resolution. This demonstrates that this approach already handles high-resolution feature maps efficiently. Furthermore, our method can train quickly on a single GPU because of this efficiency.
>
> ## Qualitative Results Prior to the CRF
>
> We have since updated the figure in Section A.3 to show both regular results and results without CRF post-processing on uncurated images. Even without CRF post-processing, our results have considerably higher resolution and fidelity than our closest baseline PiCIE. In general, the CRF’s effect is somewhat marginal: it helps to clean up some boundaries but does not change the global structure of the predictions.
>
> ## Additional Ablation Studies
>
> We added another ablation study to evaluate the effect of removing each architectural component individually. We find that their effects are still significant in these cases.
>
> ## Typos
>
> We have fixed these typos among others.
>
> Thank you again for helping us improve this work.

---

### Official Review · Reviewer_2hfj · 2021-11-02

**Correctness:** 4
**Technical Novelty And Significance:** 3
**Empirical Novelty And Significance:** 3
**Recommendation:** 6
**Confidence:** 4

**Main Review:**

The method builds on top of DINO (Caron et al., 2021). The features extracted by DINO are further refined with a small segmetation head that aims to boost performace by balancing Knn, self-correlation and random image correlation loss. The training batch consisnt of random images and random nearest neighbours (precomputed for the whole dataset). The segmentation head is a simple feed forward network - by avioding the backbone retraining (and implicitly knn recomputation), the method is very efficient to train. Hovewer, there are a nuber of problems related to the backbone - e.g., 40x40 feature resolution, which means small objects are difficult to resolve; the authors address this issue by five-cropping the dataset (corners+center crop), resulting in five times more images to find knns and better details (no rescaling to 224x224- the original transformer input size). The final steps involve clustering the resulting features and further CRF refinement for finer details in object edges. Additionally, the authors propose spatial centering on the feature correspondences, to improve segmentation of small features and clamping the segmetation correspondence at 0, to improve the optimization stability.

One does note that most performance improvement strategies are related to resolving small details that the original transformer network is incapable of doing. Therefore, I belive a future, better transformer that has fine detail capabilies would render those tricks obsolete. On the other hand, the performance is signinficantly boosted by the transformer. One would assume that future work would boost the proposed method, which has a significant performance boost over the baseline (DINO). Therefore, there are a number of plausible options of future research based on this work.

After training STEGO on Coco (ViT-small, no CRF[the code has some issues]), I also also some practical aspects that have not been raised in the paper. Looking at the initial and final confusion matrix, the performance improves on several classes and worsens on others; I believe this is mostly due to the features extracted by DINO, but sometimes the errors are amplified - no investigation on this aspect.

There is an ablation study for clamping the segmentation feature correspondence tensor at 0, five-cropping the dataset (corners+center), spatial centering on the feature correspondences and CRF. The three term loss gets no ablation study (except for SC). At the very least, I would like to see numbers with knn only, self-correlation only and random correlation only. Can we improve the performance by designing better training pairs? I believe so, and this additional ablation study would have shed light on this issue. Otherwise, we have to manually tune the $\lambda$s and the $b$ for each dataset (see page 20), and this doesn't look very appealing to me. Can the authors share some insights on this issue? I see the values are wildly different on Coco/Cityscapes. I'm willing to improve my rating, provided this issue is properly addressed.

Although well written and illustrated, there are a small number of issues with the content of the paper:
- 'typical results' images; 28 mIoU is pretty bad and most results from the paper look amazing; I have a folder full of truly random images, if anyone is interested :)
- page 5, after Eq 4, shoud read 0-clamp, not O-clamp
- page 15, In The >> In the

**Summary Of The Paper:**

The paper proposes a novel transformer-based unsupervised feature distillation method for semantic segmetation- STEGO (Self-supervised Transformer with Energy-based Graph Optimization). The pipeline has two separate stages, feature learning and cluster compactification (feature distillation). The feature network is DINO (Caron et al., 2021) - pretrained, frozen.

The main contribution is the cluster compactification(distillation) network. It features a novel 3-term loss function: Knn, self correltation and random image corelation. The authors also propose a number of strategies for performance improvement: clamping the segmentation feature correspondence tensor at 0, five-cropping the dataset (corners+center), spatial centering on the feature correspondences and conditional random fields for better segmentation at the object edges. Training the segmentation head is fast (a couple of hours on a recent GPU with the ViT-small transformer backbone) and the results are state-of-the art on semantic segmentation on CocoStuff and Cityscapes (+14mIou@CocoStuff, +7mIoU@Citiscapes).


**Summary Of The Review:**

This paper could have been outstanding, but as it stands, it's only acceptable. The best part is the unsupervised performance boost over prevoius methods (+14mIou@CocoStuff, +7mIoU@Citiscapes) and the ability to use the proposed model on top of most feature extractors. That being said, the authors have shown that the larger the transformer, the better the unsupervised segmentation, but have failed to provide a method that 'just works', regardless of the dataset. Instead, we rely on a number of tricks for performance boosting (mostly related to small objects) that could have been incorporated in the original transformer stage (fair enough, except for CRF, which is pretty much a popular solution for squeezing 2%-ish in accuracy) and some variables we need to manually set - the three-pronged loss that is at the core of the paper does not get a proper ablation study, each term is weighted empirically. Apart from the state-of-the-art performance, the training itself is kind of confusing - some classes do not benefit at all from this stage, sometimes the feature noise is happily propagated in the final segmentation, sometimes amplified - the limitations of the feature extractor and loss have not been thoroughly investigated, IMHO. This is not shown in the main paper, but I have tried the provided code on CocoStuff and those are my findings. Nevertheless, being feature extractor agnostic, future work could expand and improve the performance of this method. Furthermore, the authors suggest that label ontologies can be arbitrary - future work could benefit from a hierarchical ontology, for example.

---

> ### Author Response · Authors · 2021-11-15
> **Additional Ablation, Hyper-Parameter tuning, and Error Analysis Sections**
>
> Thank you for your thoughtful and detailed comments on our work, and for taking the time to reproduce and explore the code. We have provided a high-level overview of major additions in the general rebuttal and hope to address more targeted concerns in this comment.
>
> We have added sections to address your requests for more ablation studies, including on the three terms of our loss.
>
> We have also added a section detailing how we tune hyperparameters for each dataset and hope this addresses your concerns. We agree that hyperparameters are not ideal for any method, but argue that the majority of work in this literature has hyperparameters that control aspects of their model’s behavior and that this should not disqualify this work. We hope that future work can investigate better ways to adaptively set these hyper-parameters.
>
> You mentioned that some classes improve over the course of training while others do not improve. We appreciate you pointing us towards this phenomenon and have added a section to the appendix which explores this in greater detail. In particular, we find that many cases where STEGO amplifies the wrong class assignments, the issue is actually present in DINO’s feature correspondences. More specifically, we illustrate that the backbone’s feature correspondences predict many of STEGOs final errors which illustrates the inherent challenges that arise when the backbone representation fails to align with the label ontology. Given that the backbone is the only training signal for STEGO, it makes sense why STEGO would amplify these errors. We hope that this analysis helps clear up some of the confusion around how STEGO refines features into strong class-consistent clusters.
>
> You mentioned that we do not rescale five-cropped images to 224x224 but we wanted to clarify that we do scale images to 224x224, and this helps STEGO have an implicitly higher spatial resolution.
>
> Finally, we have corrected the typos mentioned.
>
> Thank you for helping us improve this work.

---

### Official Review · Reviewer_XoJq · 2021-11-03

**Correctness:** 4
**Technical Novelty And Significance:** 3
**Empirical Novelty And Significance:** 3
**Recommendation:** 8
**Confidence:** 4

**Main Review:**

Strengths: The authors have properly addressed related work. The paper is well-structured and the method is sound. Design decisions are backed up by ablation studies. The method is feature extractor agnostic, which still makes STEGO relevant in the future as these technologies advance.

Weaknesses: I haven't rigorously checked the mathematical formulation, but at the moment I do not find any reason to reject this paper. The pretrained backbone (DINO). used as a starting point (not fine-tuned) is quite strong and it would have been interesting to see what is the gain of using STEGO on weaker feature extractors. Also, I am a big advocate for highlighting the limitations of the method and also showing some failure cases to clearly state, from the authors' perspective, what other issues should the research community address in the future.

Minor comment: Found minor typos throughout the manuscript (difficult to pinpoint them due to the current paper format, lines without number) that do not necessarily impact the quality, but proofreading is highly recommended. (e.g. "an an", missing a/to, "O-clamp" should be "0-clamp")

**Summary Of The Paper:**

The paper introduces STEGO (Self-supervised Transformer with Energy-based Graph Optimization), a novel feature correlation refinement method that builds on top of modern self-supervised visual backbones (visual-transformers) that generate dense semantically-correlated features in an effort to improve scene semantic segmentation without any type of labels (unsupervised). Different from previous works, the authors decouple the feature learning from cluster compactification and introduce a novel contrastive loss function (a combination of three correlation factors: KNN, self and random images) in order to further constrain the features to form compact semantic clusters without damaging their consistency throughout the dataset. The authors demonstrate state-of-the-art results on two popular benchmarks CocoStuff and Cityscapes. The results are quite striking in both a quantitative (method improves segmentation by a large margin (+14mIoU and +9mIoU) compared to recently published work) and also a qualitative manner (clearly distinguish the structure of the objects in the scene).

**Summary Of The Review:**

The contribution is solid - the problem is relevant and quite difficult. There is still so much work to be done in order to close the gap between unsupervised and supervised methods for scene semantic segmentation, but nonetheless, STEGO makes a step in the right direction.

---

> ### Author Response · Authors · 2021-11-15
> **More Ablations, Limitations, and Comparisons across Backbones**
>
>
> Firstly we would like to thank the reviewer for their thoughtful comments and support of our work. We have provided a high-level overview of major additions in the general rebuttal and hope to address more targeted concerns in this comment.
>
> In our revision, we have performed additional ablation studies to justify more aspects of our loss function and to explore the performance of STEGO on the MoCoV2 feature extractor. We find that STEGO with the MoCoV2 feature extractor still beats baselines in mIoU, but does not perform as well as the DINO backbone. We hope that this addresses your comments and please let us know if you would like to see other experiments.
>
> Secondly, you mentioned having a section on the limitations of STEGO. In our original submission, we had a section detailing specific failure cases of STEGO.  We have since expanded our discussion of STEGOs limitations with an additional Section that predicts STEGOs failures from DINOs feature correspondences. This gives some insight into why some classes are not well modeled in STEGO. We also included a discussion of the challenges associated with selecting the loss hyper-parameters and hope that future work can find ways to automatically set these.
>
> Finally, we appreciate you pointing out these typos and have made another pass through the work to correct errors.
>
> Thank you again for the helpful and constructive feedback!

---

### Author Response · Authors · 2021-11-15
**General Rebuttal**

Firstly we would like to thank all of the reviewers for their detailed and thoughtful comments. In this post, We summarize our replies to address reviewers’ concerns. We have updated our manuscript and marked major changes with yellow highlights.

## More Ablations and Comparisons to MoCoV2 Backbone (2hfj, XoJq, GKTw)

All reviewers requested additional ablation studies. To this end, we have added another ablation Study (Section A.2) that measures the performance drop of removing each modification individually. This table includes ablations of the three losses (Self, KNN, and Random) at the request of reviewers 2hfj and GKTw, and a comparison to the MoCoV2 backbone as requested by reviewer  XoJq.

The ablation showed that removing each component independently harms the overall performance of STEGO, thus justifying the design decisions. We also found that the performance of our System with the ResNet MoCoV2 backbone is still significantly better in mIoU than the next best baseline (PiCIE), but this backbone performs worse than DINO.


## Additional Section on Hyper-parameter Selection (2hfj, GKTw, Xojq)

Reviewers mentioned the hyper-parameters of the method as a key challenge. To this end, we added Section A.10 which provides greater details on how to tune the parameters of STEGO even without access to labels for validation. In particular, we show that the balance between positive and negative forces can be measured by observing the distribution of feature similarities, and that good hyper-parameters yield a clear bi-modal structure. We find that this balance is the largest driver of the performance of the method. At the request of reviewers, we have also mentioned this need to tune hyper-parameters as a limitation and feel that this is a direction the community can improve on in future work.


## Deeper Analysis of STEGOs Errors (2hfj)

Reviewer 2hfj requested an analysis of why certain errors in STEGO are amplified over the course of training and Reviewer XoJq also requested additional analysis of STEGOs limitations. To address this we have added Section A.5 which shows that the correlations between features of the DINO backbone can be used to predict the kinds of errors the final STEGO architecture makes.

More specifically we show that DINO’s features tend to mix-up certain classes of the CoCoStuff ontology. Because STEGOs main source of supervision is the relationships between the backbone’s features, these errors tend to be amplified over the course of training. This shows us that one limiting factor of STEGOs performance is the quality of the backbone. Furthermore this analysis points to this, as opposed to other aspects of the training pipeline, as the root cause for many of the final errors STEGO makes.

## Qualitative analysis of the effects of the CRF (cF6T)

Reviewer cF6T requested additional qualitative results showing the effect of applying the CRF post-processing. We have since updated the figure in Section A.3 to show both regular results and results without CRF post-processing on uncurated images. Even without CRF post-processing, our results have considerably higher resolution and fidelity than our closest baseline PiCIE.

## Corrected Typos and improved clarity (cF6T)

At the request of reviewer cF6T we have moved the relationship to potts models to after the section describing the main architecture.

We thank all reviewers for all the editorial issues. We have found and corrected all typos mentioned by the reviewers and made an additional pass through the work to correct editorial issues.

---

### Author Response · Authors · 2021-11-29
**Thanks**

Thank you to all the reviewers for the helpful comments and feedback. Please let us know if there is anything else that we can address prior to the review period closing. We appreciate your help to make this work better.

---

### Public Comment · ~Evgenii_Zheltonozhskii1 · 2022-01-29
**Reproducing experemental results**

Hi,
we were trying to reproduce the experimental results presented in the paper using your code, but unfortunately instruction are a bit sparse. In particular, running with provided configuration didn't yield the results presented in the paper (we got lower results for linear evaluation and cluster collapse for clustering). Would  you mind to provide more detailed instruction for reproduction of the results (and maybe putting the code on github?

---

> ### Public Comment · ~Mark_Hamilton1 · 2022-01-31
> **Detailed Repo coming soon**
>
> Dear Evgenii, Thanks for your interest in our work. We will be releasing detailed code on github and pretrained models for our camera ready within the coming weeks. We will post the link here when ready. We appreciate the patience!

---

> > ### Public Comment · ~Evgenii_Zheltonozhskii1 · 2022-03-11
> > **Github availability**
> >
> > Hi,
> > I noted that you updated the paper with camera-ready version. Any updates about the github version of the code?

---

> > > ### Public Comment · ~Mark_Hamilton1 · 2022-03-22
> > > **Github**
> > >
> > > Hey Evgenii, thanks for your patience. We have a pre-release repo set up here. In the coming weeks we will be improving this and documenting it further:
> > >
> > > https://github.com/mhamilton723/STEGO

---

### Decision · Program_Chairs · 2022-01-20

**Decision:**

Accept (Poster)

**Comment:**

The paper received two accept and two marginally accept recommendations. All reviewers find value in the proposed supervised semantic segmentation methodology (making self-supervised representation learning towards dense prediction tasks like segmentation or clustering without explicit manual supervision) and appreciate the experimental gains, but had (mostly practical) criticism that was reasonably well addressed in the rebuttal.